

**Asian Summer Monsoon Anticyclone: Trends and Variability**

2            Ghouse Basha[1], M. Venkat Ratnam[1] and Pangaluru Kishore[2]

[1]National Atmospheric Research Laboratory, Department of Space, Gadanki-517112, India.

4       [2] Department of Earth System Science, University of California, Irvine, CA, 92697, USA.

5            Correspondence to: Ghouse Basha (mdbasha@narl.gov.in)

**Abstract**

7            The Asian Summer Monsoon (ASM) dynamics act as a pathway for the transport of

trace gases and pollutants both vertically (through convection) and horizontally (through low-
level jet and tropical easterly jet). These pollutants will be trapped in the anticyclone present
during the same period in the upper troposphere and lower stratosphere (UTLS). Since the
anticyclone extends from the Middle East to East Asia, trapped pollutants are expected to
make a large radiative forcing to the background atmosphere. Thus, it is essential to
understand the anticyclone features in detail and its relation to long-term oscillations. This
work explores the spatial variability and the trends of the Asian Summer Monsoon
Anticyclone (ASMA) using observational and reanalysis data sets. Emphasis is made to
investigate the temporal, spatial, and long-term trends of ASMA. Our analysis indicates that
the spatial extent and magnitude of ASMA is greater during July and August compared to
June and September. The decadal variability of the anticyclone is very large at the edges of
anticyclone than at the core region. Significant deviations in the northeast and southwest parts
of ASMA are also observed in the decadal variability with reference to 1951-1960 period.
The strength of the ASMA shows a drastic increase from the easterlies to the westerlies in
terms of temporal variation. Further, our results show that the extent of anticyclone is greater
during the active phase of the monsoon, strong monsoon years, and during La Niña events.
Significant warming with strong westerlies is observed exactly over the Tibetan Plateau
during the active phase of the monsoon, strong monsoon years, and during La Niña events.
Over the Tibetan Plateau, there is strong elevated heating from the surface to the tropopause,



which is observed with strong westerlies during active and strong monsoon years. Our results
support the transport process over Tibetan Plateau and the Indian region during active, strong
monsoon years and during strong La Niña years. It is suggested to consider different phases
of monsoon while interpreting the pollutants/trace gases in the anticyclone.
*Keywords*: Asian Monsoon, anticyclone, geopotential height, La Niño, El Niño, and rainfall
**1. Introduction**
The Asian Summer Monsoon Anticyclone (ASMA) is a dominant circulation in the
Northern Hemisphere (NH) summer in the Upper Troposphere and Lower Stratosphere
(UTLS), which extends from Asia to the Middle East. ASMA is bordered by the subtropical
westerly jet in the north and easterly jets to the south. ASMA circulation responds to heating
corresponding to the deep convection of the south Asian monsoon (Hoskins and Rodwell,
1995; Highwood and Hoskins, 1998). This strong anticyclone circulation isolates the air and
is tied to the outflow of deep convection, which has distant maxima characters in terms of
dynamical variability and chemical characteristics (Randel and Park, 2006; Park et al., 2007).
The maximum occurs due to strong winds and closed streamlines of an anticyclone, which
isolate the air within the anticyclone and it is very dynamic in nature (e.g. Vogel et al., 2016).
Recently, the anticyclone circulation in UTLS has been paid more attention by
researchers in order to understand dynamics, chemistry and radiation of the region. The
dynamical confinement of tropospheric tracers and aerosols in the anticyclone isolate them
from the surrounding air displaying distinct maxima near the tropopause. This issue has been
discussed by several authors (e.g., Park et al., 2007; Fadnavis et al., 2014; Glatthor et al.,
2015; Vernier et al., 2015; Santee et al., 2017). Deep convection during monsoon can
transport tropospheric tracers from the surface to the UTLS (Vogel et al., 2015; Tissier and
Legras, 2016). The confined tracers transported outside the edge of the anticyclone will affect
the trace gas concentration in the UTLS resulting in significant changes in radiative forcings



(Solomon et al., 2010; Riese et al., 2012; Hossaini et al., 2015). The centre of the anticyclone
is located either over the Iranian Plateau or over the Tibetan Plateaus where the distribution
of pollutants and tracers vary significantly (Yan et al., 2011).

The spatial extent, strength, and the location of an anticyclone vary on several

temporal scales caused by internal dynamical variability of the Asian monsoon (Zhang et al.,
2002; Randel and Park, 2006; Garny and Randel, 2013; Vogel et al., 2015; Pan et al., 2016).
However, the variability of anticyclone structure and response to Indian monsoon are not
understood. Therefore, in the first part of the study, we investigate the spatial, inter-annual
and decadal variations of the anticyclone. Since the Indian monsoon responds in different
time scales, we also investigated the anticyclone variability with respect to the wet and dry
spells of the Indian monsoon, strong and weak monsoon years, and the stronger El Nino
Southern Oscillation (ENSO) years. For this, we have utilized the NCEP/NCAR reanalysis
geopotential height from 1948 to 2016. The structure of the paper is as follows. We describe
the data sets used in this study in Section 2. Section 3 contains the seasonal and decadal
variation of the anticyclone and its relation with large scale oscillations. Section 4 shows the
influence of days with wet and dry spells, strong and weak monsoon years, and ENSO's
effects on the anticyclone. Finally, we discuss our results in Section 5.
**2. Data and Methodology**
**2.1. NCEP/NCAR Reanalysis**

The National Centers for Environmental Prediction (NCEP), in collaboration with the

National Center for Atmospheric Research (NCAR) produces reanalysis data from a
consistent assimilation and modeling procedure that incorporates all the available observed
conditions obtained from conventional and satellite information from 1948 to the present
(Kalnay et al. 1996). We used NCEP/NCAR reanalysis daily geopotential height (GPH) and
wind data from the years 1948 to 2016. The NCEP/NCAR data assimilation uses a 3D-



variational analysis scheme with 28 pressure levels and triangular truncation of 62 waves
(horizontal resolution of 200m). Both GPH and temperature at the chosen standard levels are
described as class output variables (Kalnay et al. 1996) i.e. they are strongly influenced by
observed data. Only the Indian summer monsoon months (June, July, and August,
September) containing gridded daily data were considered in this study. The NCEP/NCAR
reanalysis data had a spatial resolution of 2.5°. The seasonal values are estimated from daily
data. To identify the spatial and temporal variations of the anticyclone centres, we used the
monthly mean values of the GPH and the zonal wind component. The quality of NCEP GPH
reanalysis data can be found from Bromwich et al., (2007).
**2.2. IMD Gridded Precipitation Data**

The India Meteorological Department (IMD) high-resolution ($0.25^o$x$0.25^o$) gridded

precipitation data is used to identify the wet and dry spells during June, July and August
months from 1901-2016. This precipitation data has been validated extensively with
observational and reanalysis data sets and displays very good correlation (Kishore et al.,
2016). For identification of active (or wet) and break (or dry) spells, we followed the similar
procedure as described by Rajeevan et al. (2010) and Pai et al. (2016) over the monsoon core
zone ($18^o$N-$28^o$N, and $65^o$E-$88^o$E). Data from 1948-2016 have been used.
**2.3. GNSS Radio Occultation (RO) Data**

We also used the Global Navigation Satellite System (GNSS) RO data for investigating

the temperature anomaly. The basic measurement principle of RO exploits the atmosphere-
induced phase delay in the GNSS signals, which are recorded in the low earth orbiting
satellite. This technique provides vertical profiles of refractivity, density, pressure,
temperature, and water vapour (Kursinski et al., 1997). The temperature profiles from this
technique are available with low horizontal (~200-300 km) and high vertical resolutions (0.5-
15 km) with accuracy of <0.5 K. We used the CHAllenging Minisatellite Payload (CHAMP)



and Constellation Observing System for Meteorology, Ionosphere, and Climate (COSMIC)
covering the period from 2002 to 2016.

The CHAMP satellite was launched on 15 July 2000 in to a circular orbit by Germany

to measure the Earth's gravity and magnetic field and to provide global RO soundings
(Wickert et al. 2001). About ~230 RO profiles per day were measured by the CHAMP
payload since 2002. The CHAMP payload was solely designed to track the setting
occultations, and the RO event gets terminated when the signal is lost, which results in a
decrease in the number of occultations with a decreasing altitude (Beyerle et al. 2006). This
receiver measures the phase delay of radio wave signals that are occulted by the Earth's
atmosphere. From this phase delay, it is possible to retrieve the bending angle and refractivity
vertical profiles. The CHAMP data was available from 19 May 2001 to 5 October 2009.

COSMIC consists of a constellation of 6 satellites, which was launched in April 2006

to a circular, 72° inclination orbit at a 512 km altitude capable of receiving signals from the
Global Positioning System (GPS) (Anthes et al., 2008). Compared to previous satellites,
COSMIC satellites employed an open loop mode, which can track both the rising and setting
of occultations (Schreiner et al. 2007). The open-loop tracking technique significantly
reduces the GPS RO inversion biases by eliminating tracking errors (Sokolovskiy et al.
2006). The COSMIC temperature profiles display a very good agreement with radiosonde
data, reanalyses, and models (Rao et al., 2009; Kishore et al., 2011; Kishore et al., 2016). We
derived the cold point tropopause altitude/temperature over the ASMA region as discussed by
Ratnam et al. (2014) and Ravindrababu et al. (2015). Both the CHAMP and COSMIC data
were obtained from COSMIC Data Analysis and Archive Center (CDAAC) (https://cdaac-
www.cosmic.ucar.edu/cdaac/products.html).

**3. Results and Discussion**



### 3.1. Variability of the Anticyclone

Climatological spatial variability of the GPH along wind vectors at 100 hPa during June, July, August and September months from NCEP reanalysis data is shown in Figure 1(a, b, c & d). The anticyclone circulation is clearly depicted during June, July, August and September by wind vectors (Figure 1). During the month of September and June the values of GPH are low compared to July and August which represents the spatial extent of the anticyclone. The spatial extent and intensity of anticyclone are greater during July compared to the intensities present during other months. During July and August, the anticyclone extends from the Middle East to East Asia. The spatial extent of anticyclone circulation is clearly evident in the grid $15^{o}$N-$45^{o}$N; $30^{o}$E-$120^{o}$E at 100 hPa level and the climatological averaged values of GPH varies from 16.5-17 km in NCEP reanalysis during 1948-2016. Using the modified potential vorticity equation, Randel et al. (2006) showed the spatial variation of anticyclone where GPH values are stationary in the range of 16.75-16.9 km. Similarly, Park et al. (2007) showed the anticyclone structure from the strongest wind at 100 hPa through streamline function. Bian et al. (2012) reported the spatial variability of anticyclone using 16.77 km and 16.90 km in the GPH contour as the lower and the upper boundary, respectively. Thus, these empirically selected GPH values represent anticyclone boundaries. Therefore, in this present study, we have chosen the values from 16.75 to 16.9 km to investigate the spatial features of the anticyclone and the resultant picture is depicted in Figure 1(e, f, g & h). The spatial extent and existence of anticyclone is highly prominent during July and August compared to June. During the September month, the GPH values in the range 16.75-16.9 km are not present. Therefore, we considered the average of July and August GPH from 1948-2016 for further analysis as shown in Figure S1. The core region and the spatial extent of the anticyclone are clearly evident from Figure S1. The core region of anticyclone shows bimodal distribution i.e. one core located at the south-western flank of the Himalayas and another over Iran. The



core region over the south-western flank of Himalayas is due to large scale updraft, which is
caused by the moist energy over Indo-Gangetic plain, heating of Tibetan plateau, and the
orographic forcing of the Himalayas. Severe heating over Arabian Peninsula supports the
formation of the mid-tropospheric anticyclone in the west. This anticyclone can merge
intermittently within ASMA. It is also observed that the spatial extent of anticyclone varies
drastically at different temporal scales. Therefore, seasonal variation is much more
pronounced.

The decadal variation of the anticyclone is studied with respect to the spatial

variability. Figure 2 shows the decadal spatial variation of the anticyclone with reference to
the years 1951-1960. The significant difference in the decadal variation is noticed from
Figure 2. The edges (east, north, and west) of the anticyclone undergo drastic changes during
the period 1961-1970. In case of 1971-1980 period, except for a small portion in the east, the
whole anticyclone shows drastic changes. During the decade 1971-1980, the recorded GPH
values in anticyclone are lower by ~ 25 m when compared to the values in 1951-1960. This
feature is quite opposite during 1981-1990 where high values (~30 m) are observed compared
to those in the reference period. The GPH difference is significant over the west, northeast
and southern regions of the anticyclone during the 1991-2000 period. Similar changes are
observed during 2001-2010. Compared to all the decadal differences, 2011-2016 shows a
completely different picture. The changes are only in the western and north-eastern corner,
whereas other parts of the anticyclone do not show any change. From this analysis, we
observed significant changes in the anticyclone even from one decade to another, which can
result in a change in chemical and dynamical changes over this region.

Further, the spatial distribution of trends is estimated during the years 1948-2016 by

using robust regression analysis and is displayed in Figure 3. Spatially, the anticyclone trend
shows two distinct pictures. The edges on all side of the anticyclone undergo noticeable



changes compared to the core region. The northern side of the anticyclone shows reduction
(~30 m) in the strength whereas the southern part illustrates increases in strength. Therefore,
in order to understand the asymmetry in the anticyclone variability, we have divided the
anticyclone region into 4 different sectors as shown in Figure 4 based on the peak values of
GPH along longitude and latitude cross-sections. The center values of GPH are located at
$70^{o}$E longitude and $32.5^{o}$N Latitude. The four sectors can be divided as South-East (SE)
($22.5^{o}$N-$32.5^{o}$N), North-East (NE) ($32.5^{o}$N-$40^{o}$N) in the longitude band of $70^{o}$E-$120^{o}$E,
South-West (SW) ($22.5^{o}$N-$32.5^{o}$N), and North-West (NW) ($32.5^{o}$N-$40^{o}$N) at the $20^{o}$E-$70^{o}$E
longitude range. The average time series (July and August) of zonal wind anomalies in these
sectors from 1948-2016 are shown in Figure 5. The zonal wind shows a clearly increasing
trend in all the sectors. From 1948 to 1980 the zonal wind anomalies are easterlies and later
on, clear shift is noticed towards the westerlies. This represents that the westerlies are more
dominant in recent decades with a strong increase in magnitude. The change is highly
significant in the north-west and north-east sectors with a magnitude variability of 10 m/s
from 1948-2016 whereas it is 5 m/s in the south-east and south-west sectors. It is to be noted
that the winds in the anticyclone will not be contaminated with the tropical easterly jet
persisting during the monsoon season as their cores are well separated. One is located in the
northern part and the other in the southern part of India. In addition, we estimated the strength
of the anticyclone during the monsoon season by using a difference in the zonal wind
between the northern (30°N-40°N) and southern (10°N-20°N) flanks of the anticyclone,
which is depicted in Figure 5e. A significant increase in the strength of the anticyclone is
noticed from Figure 5e at a rate of 0.184 m/s per year (12 m/s from 1948-2018).

It is well known that the Indian monsoon rainfall varies at different time scales i.e.

daily, sub-seasonal, interannual, decadal and centennial scales. Precipitation during the
monsoon varies from intra-seasonal scales between active (good rainfall) and break (little





rainfall) spells. Any small change in the precipitation pattern will affect the anticyclone due
to the thermodynamics involved in rainfall. In this study, we also investigated the anticyclone
variability (during the active and break spells of the Indian monsoon. The active and break
spells were identified in July and August by using the high resolution gridded ($0.25^o$ x $0.25^o$)
rainfall data from 1948 to 2016 as defined by Pai et al. (2010).

The number of active and break days is derived from the precipitation data shown in

Figure S2 (a & b). Daily GPH, temperature, and zonal wind are taken from NCEP reanalysis
whereas the tropopause altitude is derived from the GNSS RO data for active and break days.
The anticyclone structure during active (red line) and break (blue line) days are shown in
Figure 6a. Two interesting aspects of the anticyclone variability can be noticed between
active and break days. One aspect is the extent  of the anticyclone is large during active days
compared to break days and another is the existence of two cell structures in the anticyclone
core region during active days. The extent is large in the eastern and northern side in active
days. The zonal (meridional) cross section of temperature (color shade), zonal wind (contour
lines) difference between active and break phase averaged in the longitude band of $80^o$E-$90^o$E
(latitude band of $30^o$N-$40^o$N) along with cold point tropopause for active and break days is
illustrated in Figures 6b & 6c. During active days, temperature shows cooling in tropical
latitudes whereas it shows warming in the mid-latitudes from surface to tropopause.
Significant warming is observed during the active days in the mid-troposphere over the
Tibetan Plateau and its northern side. Westerly (easterly) winds exist over the warmer and
cooler regions. The warm temperature anomalies stretch from 1.5 to 12 km in between $25^o$N
and $60^o$N. The tropopause altitude is low (high) during the active (break) phase of Indian
monsoon as show in Figure 6b. The meridional cross-section of temperature anomalies
displays significant warming from ~1.5 to 8 km over the Indian region. The tropopause
altitude exemplifies random variability in the meridional cross section.



As discussed previously, the anticyclone circulation is significant during the months
of July and August when most of the precipitation occurs over India (Basha et al., 2015;
Kishore et al., 2015). The influence of strong and weak monsoon years will have a drastic
impact on anticyclone circulation. In order to understand these changes, we have divided the
years into strong and weak monsoon years based on gridded precipitation data over the
domain 5$^{o}$N-30$^{o}$N and 70$^{o}$E-95$^{o}$E from the years 1948-2016. This region is known to have
heavy precipitation and orographic forcing, which helps transport of water vapour through
deep convection to UTLS (Houze et al., 2007; Medina et al., 2010; Pan et al., 2016). The
detrended precipitation represents the strong and weak monsoon years. Years with positive
(negative) values of precipitation shows the strong (weak) monsoon years as shown in Figure
S2b. Further, we have divided the GPH, temperature at 100 hPa tropopause altitude based on
strong and weak monsoon years. The composite of mean distribution of anticyclone
circulation during strong and weak monsoon years is shown in Figure 7a. The circulation
expands on the eastern and western sides of the anticyclone during the weak monsoon (blue
line) years. The core of the anticyclone is significant during strong monsoon years. Clear eye
structure is observed on the right (left) side of the anticyclone in the core region during the
strong (weak) monsoon years. The composite mean difference of temperature and zonal wind
between the strong and weak monsoon years along with tropopause altitude averaged in the
longitude range of 80-85$^{o}$E is shown in Figure 7b. The warmest temperature anomalies are
observed over the Tibetan Plateau. Positive (warm) temperature anomalies exactly above the
Tibetan Plateau (11 km) and negative (cooling) on both sides are noticed in the lower
troposphere from Figure 7b. Strong easterlies (westerlies) winds are observed on the left
(right) side of the Tibetan Plateau. The whole Tibetan Plateau acts as a barrier that drives the
cold air to upper altitudes during strong monsoon years. Strong anticyclone circulation with
strong westerlies at 35$^{o}$N and easterlies on both sides with elevated tropopause represent the



impacts of the strong monsoon vertically above the anticyclone. The raising motion over East
Asia excited by the local heating of the Tibetan Plateau links to the single stretch vertically.
The longitude and altitude cross-section of temperature and wind anomalies shown in Figure
7c are averaged between a latitude band of 35-40°N. Positive temperature anomalies are
observed from the surface to 12 km in the longitudes 60-80°E and stretch towards the west.
This clearly demonstrates that a large scale ascent develops over the Asian monsoon region.
The tropopause altitude is high (low) during strong vertical motion and heavy precipitation
found over the region similar to that reported by Lau et al. (2018). The transport processes
from the boundary layer to the tropopause occurs on the east side of the anticyclone i.e.
southern flank of Tibetan Plateau, northeast India and the head of the Bay of Bengal. This
result is consistent with the previous studies by Bergman et al. (2013).

ENSO typically shows the strongest signal in boreal winter, but it can affect the

atmospheric circulation and constituent distributions until the next autumn.  It is well-known
that strong ENSO events have a significant influence on tropical upwelling and STE. This
change can impact the distribution of composition and structure of UTLS region. In the
UTLS region, the tropopause responds to the annual and interannual variability associated
with ENSO (Trenberth, 1990) and QBO (Baldwin et al., 2001). Several studies have been
focused on the effects of the different impacts of El Niño on tropopause and lower
stratosphere (Hu and Pan, 2009; Zubiaurre and Calvo, 2012; Xie et al., 2012).  In the present
study, we have investigated the changes associated with strong ENSO events with the
anticyclone circulation and tropical upwelling during July and August. Therefore, we have
also separated the GPH for the strongest El Niño (1958, 1966, 1973, 1983, 1988, 1992, 1998,
and 2015) and La Niña (1974, 1976, 1989, 1999, 2000, 2008, and 2011) years to verify the
change in the circulation pattern of the anticyclone. For this we have chosen July and August
GPH data at 100hPa as shown in Figure 8. The red and blue colors indicate the composite of



the La Niña and El Niño circulation. During the El Niño, the anticyclone circulation is
stronger and extends over the La Niña at 100 hPa as shown in the Figure 8a. On the eastern
and southern sides of the anticyclone, the expansion is more during the La Niña years. The
warm temperature with strong westerlies in the latitude band of $43^oN$-$55^oN$ is observed
during the El Niño as shown in Figure 8b (Lau et al., 2018). The cooling impact is significant
over the Tibetan Plateau during La Niña events compared to El Niño events. Significant
cooling is observed over the Tibetan Plateau and distributes towards tropical latitudes
between 600-100 hPa. The zonal wind shows a convergence of easterly winds over the
Tibetan Plateau from the mid to the upper tropospheric region. In the right side of the Tibetan
Plateau there exist strong westerly winds from the surface to the tropopause altitudes with
strong warming. The meridional cross-section of temperature and the zonal wind difference
between La Niña and El Niño is shown in Figure 8c. Significant cooling is observed during
La Niña in the longitude band of $80^oE$-$100^oE$ with strong easterlies from the surface to the
tropopause. From this analysis, it is clear that the Indian summer monsoon variability has a
significant impact on ASMA, and it is necessary to consider the different phases of monsoon
while dealing with UTLS pollutants. In addition, we have investigated the zonal mean
vertical cross-section in the longitude band of $50$-$60^oE$,which represents the Iranian Mode.
Figure S3 depicts the difference between active and break phases, strong and weak monsoon
years, and La Niño and El Niño years along with the tropopause altitude. Significant warming
is observed during La Niña years and strong monsoon years compared to the active phase of
the Indian monsoon in the troposphere. Compared to the Tibetan mode, the Iranian mode
warming is less. The tropopause altitude is slightly higher during the active phase of the
Indian monsoon, strong monsoon years and La Niña years. A moderate increase in
tropopause from equator to $40^oN$ is observed and decreases drastically afterwards.
**4.  Summary and Conclusions**



Several authors discussed the interannual and decadal variability of pollutants and tracers
in the ASMA region from the model,observational and reanalysis data sets (Kunze et al.,
2016; Santee et al., 2017; Yuan et al., 2019). In this present study, we have investigated the
spatial variability, trends of the anticyclone and the influence of Indian monsoon activity i.e.
active and break days, strong and weak monsoon years, and strong La Niña and El Niño years
on ASMA using long-term reanalysis and observational data sets that were not investigated
earlier. We have considered the GPH values from 16.75 km to 16.9 km, which represents the
spatial structure of anticyclone at 100 hPa in this study. Our analysis shows that the spatial
(magnitude) of the anticyclone structure is very large (strong) during July followed by
August whereas it is very weak in June at 100 hPa. The bimodal distribution (Tibetan and
Iranian modes) of the anticyclone is clearly observed during the month of July which is not
present during other months (June and August). The anticyclone variability undergoes
significant decadal variations from one decade to another. The edges of ASMA changes
drastically compared to the core of anticyclone. However, there are significant spatial
differences in the structure of the anticyclone at 100 hPa. The anticyclone undergoes a
decreasing trend on the northern side whereas an increasing trend on the western part. A
significant increasing trend is observed in the spatially averaged zonal wind in four different
sectors (Figure 5). The zonal wind anomalies illustrate easterlies from 1948 to 1980 and
westerlies thereafter. In the recent decade the westerlies are significant in the anticyclone
region at 100 hPa. The change is significant in the north-western and north-eastern sectors
with a magnitude variability of 10 m/s from 1948-2016 whereas it is 5 m/s in the south-
eastern and south-western sectors. The strength of the anticyclone increases with a rate of
0.184 m/s per year (12 m/s from 1948-2016) in the anticyclone region (Figure 5e). Yuan et al.
(2019) also reported the increasing trend in the strength of the anticyclone by considering the
MERRA 2 reanalysis data from 2001-2015.



Further, we have investigated the Indian monsoon influence on the anticyclone region.
Our results reveal that the spatial extent of the anticyclone expands during the active phase of
the Indian monsoon, the strong monsoon years and during strong La Niña years on the
northern and eastern sides. During these events, the bimodal distribution (Tibetan and Iranian
modes) of the anticyclone is noticed. A similar expansion of the anticyclone is noticed during
strong monsoon years from MERRA2 data by Yuan et al. (2019). However, the ASMA
boundaries are not always well defined in all the events. The zonal mean cross-section of
temperature shows significant warming over the Tibetan Plateau and from the surface to 12
km during the active phase of the Indian monsoon, the strong monsoon years, and the strong
La Niña years. Similarly, the rise of tropopause during the active phase of the Indian
monsoon, the strong monsoon years and the strong La Niña years is noticed. Since the
Tibetan Plateau acts as a strong heat source in summer with the strongest heating layer lying
in the lower layers, the thermal adaptation results in a shallow and weak cyclonic circulation
near the surface, and a deep and strong anti-cyclonic circulation above it. During summer, the
Tibetan Plateau acts as a strong heat source, which influences the whole UTLS region. The
warm ascending air above will pull the air from below; the surrounding air in the lower
troposphere converges towards the Tibetan Plateau area and climbs up the heating sloping
surfaces (Bergman et al., 2013; Garny and Randel, 2016). Significant warming is observed
over the Tibetan Plateau, which represents the strong transport of pollutants into the
tropopause during the active phase of the Indian monsoon, the strong monsoon years, and the
strong La Niña years. Pan et al. (2016) reported the transport of carbon monoxide through the
southern flank of the Tibetan Plateau from the model analysis. The above mentioned results
indicate that the high mountain regions play a significant role in elevated heat sources during
the formation and maintenance of the anticyclones over Asia. It emphasizes the role of the
thermal forcing of the Tibetan Plateau on the temporal and the spatial evolution of the South



Asian High. Lau et al. (2018) showed that the transport of the dust and pollutants from the
Himalayas-Gangetic Plain and the Sichuan Basin.
Overall, we demonstrate the ASMA variability during different phases of the Indian
monsoon. The uplifting of boundary layer pollutants to the tropopause level occurs primarily
on the eastern side of the anticyclone, centered near the southern flank of the Tibetan Plateau,
north-eastern India, Nepal, and north of the Bay of Bengal. However, a more detailed and a
higher quality of dataset is needed to further understand the effects of the Tibetan Plateau on
the transport of different tracers and pollutants to the UTLS region (Ravindrababu et al.,

2019).


*Data Availability*. The NCEP/NCAR reanalysis data are available from NOAA website
(https://www.esrl.noaa.gov/psd/data/gridded/data.ncep.reanalysis.pressure.html).     The
COSMIC and CHAMP data is available from COSMIC CDAAC website.   IMD gridded
precipitation data is available at National Climate data center Pune, India. All the data used in
the present study is available freely from the respective websites.
*Authors'Contributions*. GB and MVR conceived and designed the scientific questions
investigated in the study. GB performed the analysis and wrote the draft in close cooperation
with MVR. PK estimated the active and break spells of the Indian monsoon. All authors
edited the paper.
*Competing Interests*. The authors declare that they have no competing financial interests.
*Acknowledgements*. We thank NCEP/NCAR reanalysis for providing reanalysis data. We
thank CDAAC for production of COSMIC and CHAMP GPSRO data and IMD gridded
precipitation data from National Climate data center Pune, India. This work was supported by
National Atmospheric Research Laboratory, Department of Space, and India

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



**Figures**

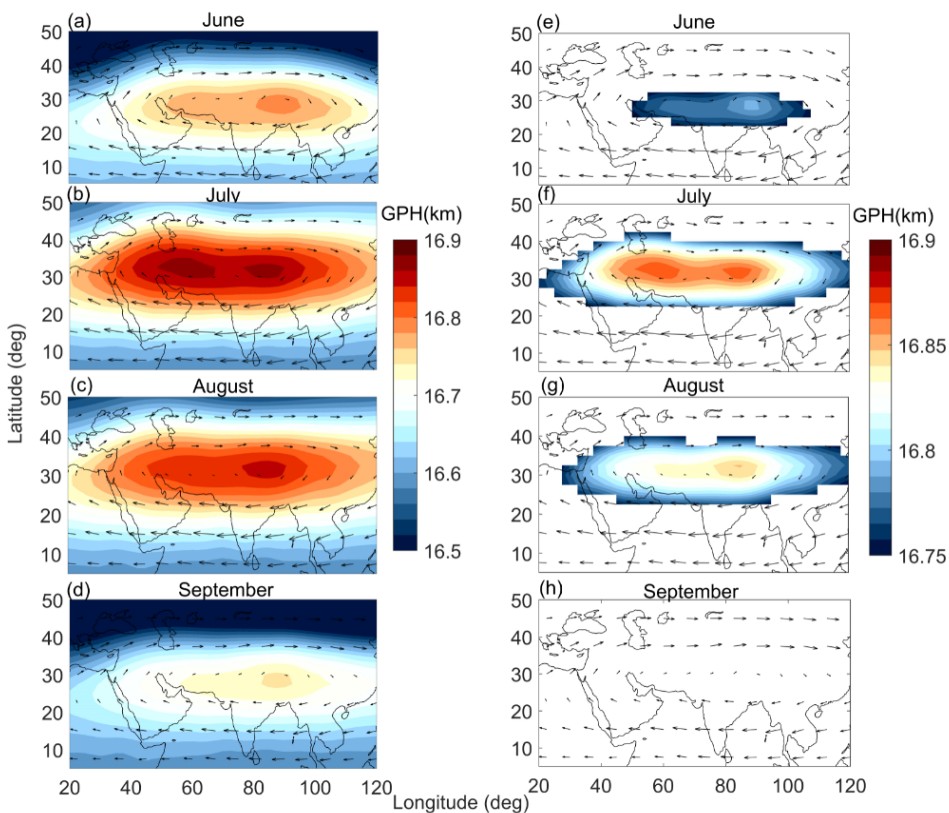


Figure 1. Spatial distribution of Geopotential Height (GPH) and wind vectors at 100 hPa
during (a) June, (b), July, (c) August and (d) September from NCEP reanalysis data
averaged from the year 1948-2016.  The core of the anticyclone region was chosen based
on the GPH values ranging from 16.75 to 16.9 km. The spatial extent and magnitude of the
anticyclone after applying the GPH criteria for (e) June, (f) July, (g) August and (h),
September.





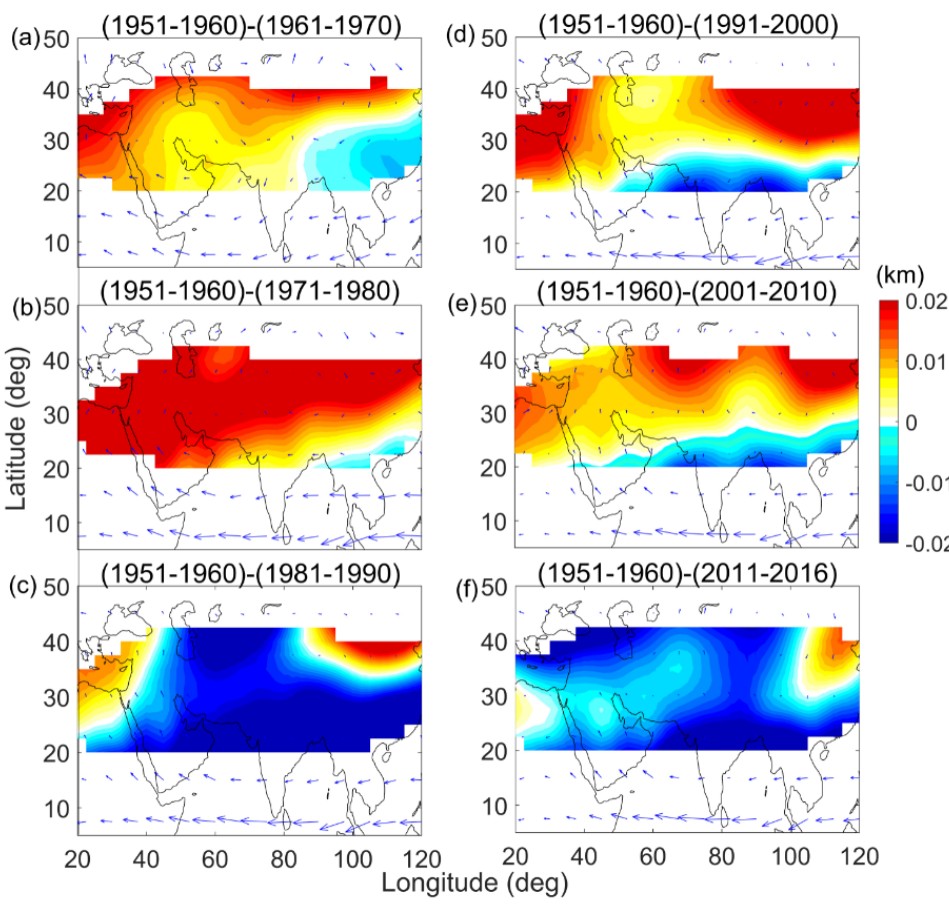


Figure 2. Decadal variation of anticyclone obtained from GPH and wind vectors with

reference to 1951-1960 period.










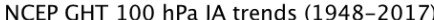

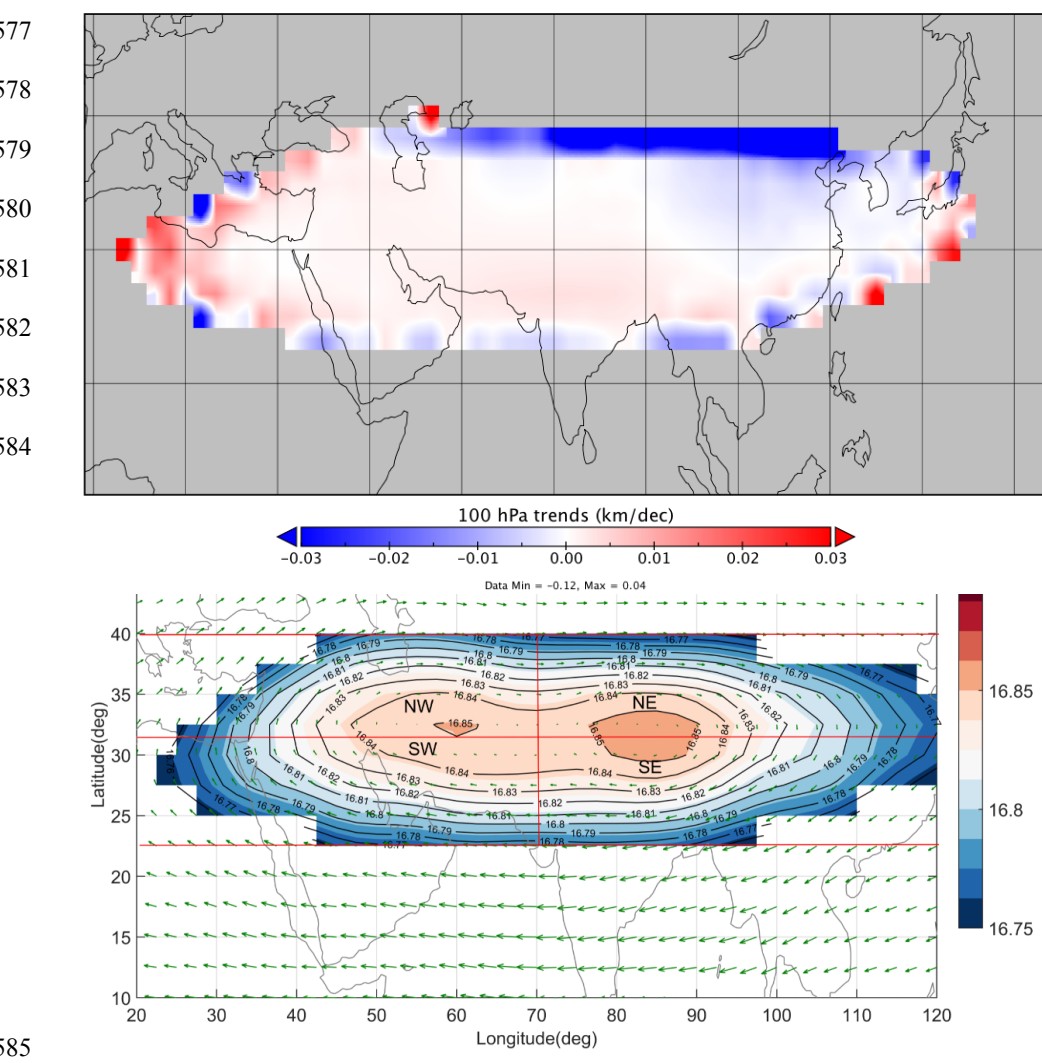



Figure 4. The climatological distribution of GPH (16.75 to 16.9 km) and wind vectors

averaged during July and August from NCEP reanalysis data along with contour lines at

100 hPa. The anticyclone region is further divided in to 4 sectors based on peak values of

GPH. The GPH values peak centres at 32.5° N in latitude and 70°E in longitude. The

sectors are further divided in to South-East (SE) (22.5°N-32.5°N), North-East (NE)

(32.5°N-40°N) in longitude band 70°E-120°E, South-West (SW) (22.5°N-32.5°N), and

North-West (NW) (32.5°N-40°N) at 20°E-70°E longitude range.



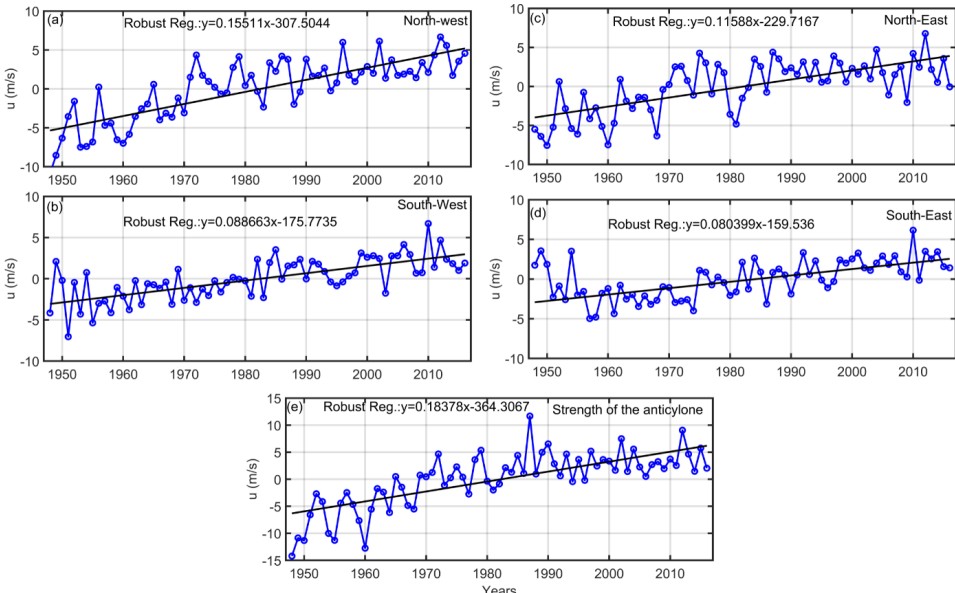

Figure 5.Time series of anomalies in zonal wind estimated for (a) North-West, (b) South-West, (c) North-East and (d) South-East sectors of ASMA. The trend analysis was performed at 95% confidence interval by using robust regression analysis. (e) The strength of the anticyclone was estimated from the zonal wind difference between (30°N-40°N)-(10°N-20°N) in the longitude band of 50°E-90°E.

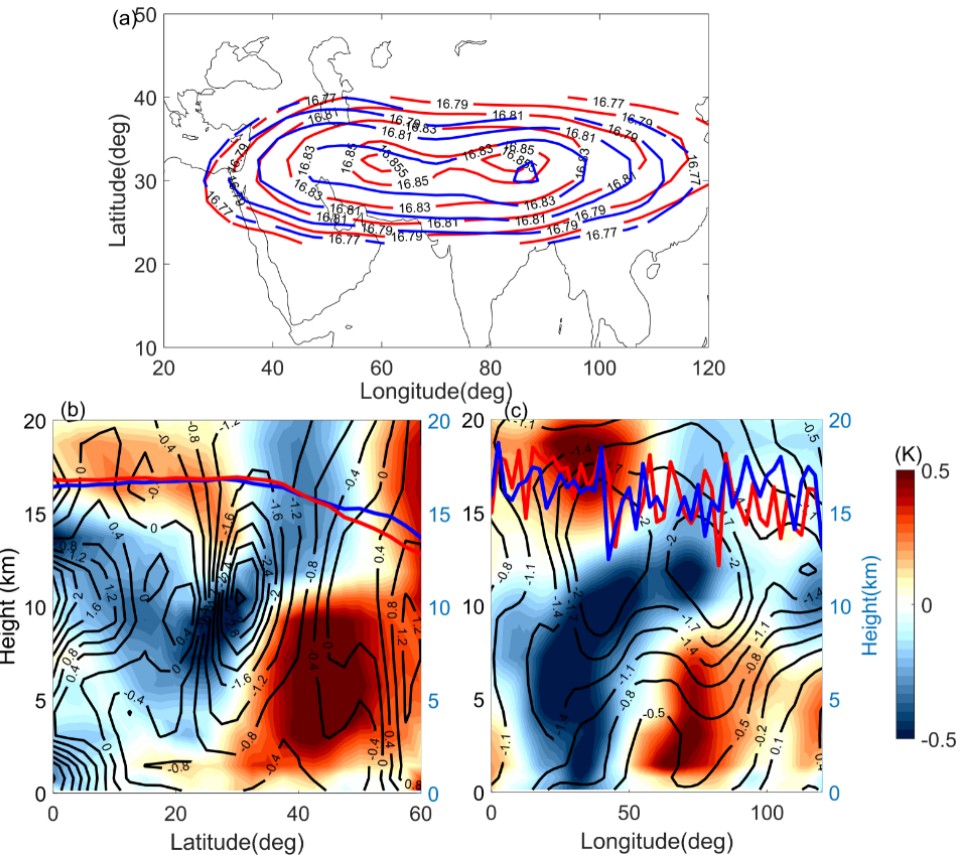


Figure 6. (a) ASMA variability during active and break phases of Indian monsoon obtained
from GPH at 100 hPa. Red line indicates the active and blue line for break phase of Indian
monsoon. (b) Latitude-altitude cross-section of temperature (colour shaded, K) and zonal
wind anomalies (contour lines, m/s) which are estimated from difference between active
and break phases of Indian Monsoon in the longitude band of 80°E-90°E. (c) Longitude-
altitude cross-section of temperature and wind anomalies averaged between 30°N-40°N.
The red and blue lines in Figure 6b & 6c denotes the tropopause altitude during active and
break spells of Indian monsoon estimated using GNSS RO data.


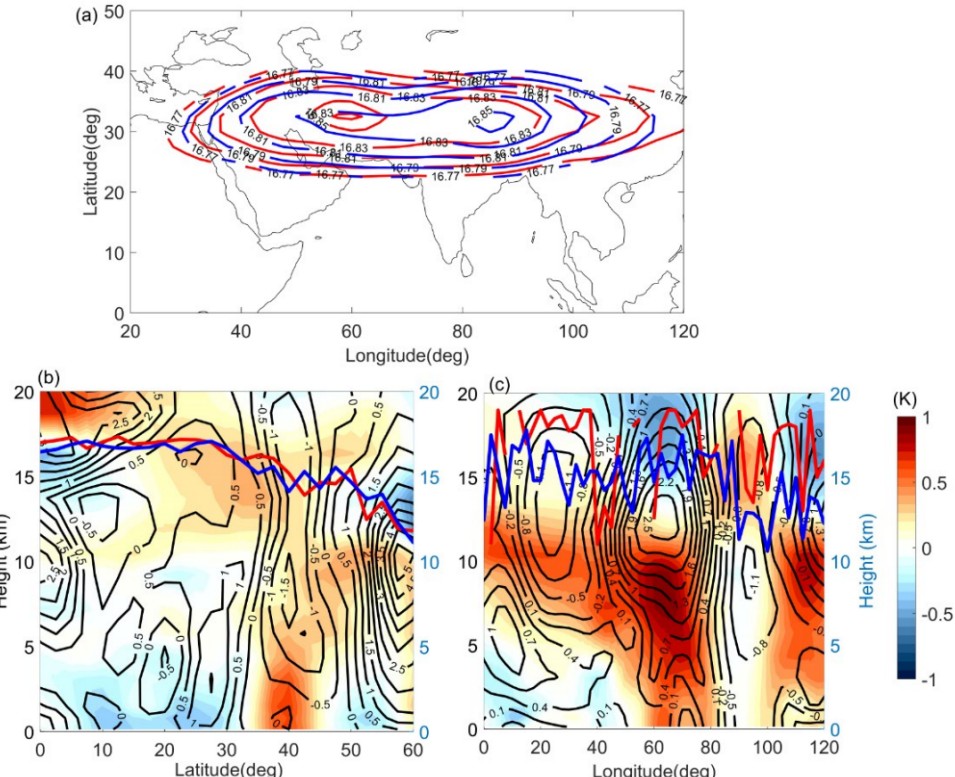



Figure 7. (a) ASMA variability obtained from GPH at 100hPa during strong and weak

monsoon years calculated based on high resolution rainfall data in band of 5°N-30°N,

70°N-95°E grid. Red line indicates the strong and blue line for weak monsoon years. (b)

Latitude-altitude cross-section of temperature (colour shaded, K) and zonal wind

anomalies (contour lines, m/s) which are estimated from difference between strong and

weak monsoon years in the longitude band of 80°E-90°E. (c) Longitude-altitude cross-

section of temperature and wind anomalies averaged between 30°N-40°N. Red and blue

lines in Figure 7b & 7c denote the tropopause altitude during strong and weak monsoon

620        years estimated using GNSS RO data.



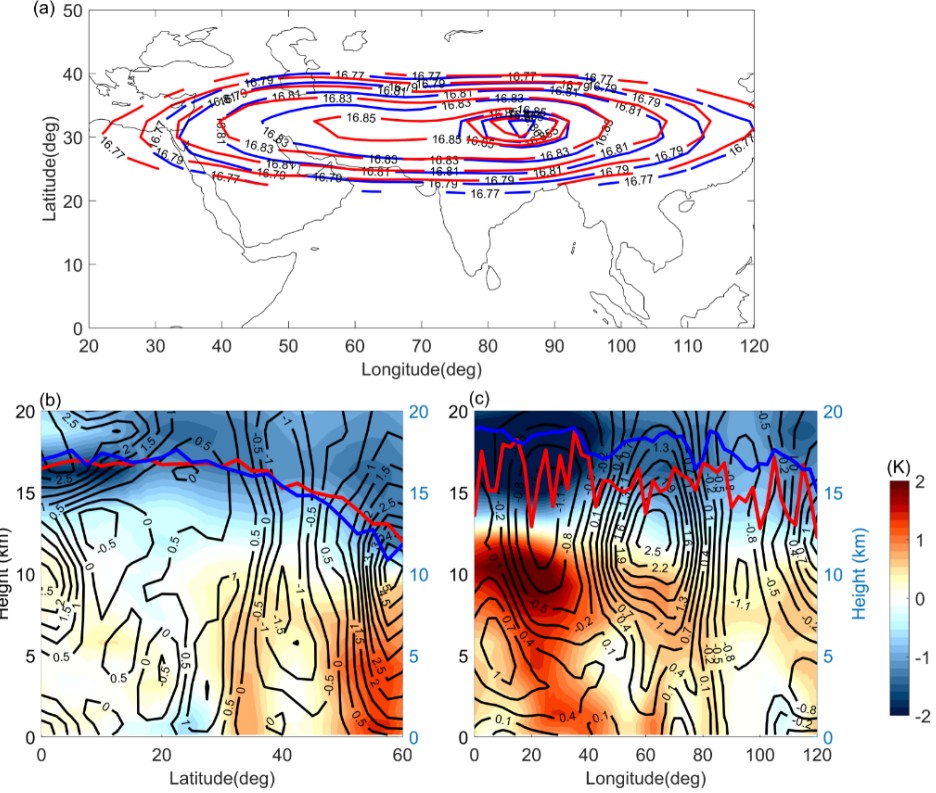


Figure 8. (a) ASMA variability obtained from GPH at 100 hPa during strong La Niño and El

Niño years. Red and blue lines indicate the La Niño and El Niño years. (b) Latitude-

altitude cross-section of temperature (colour shaded, K) and zonal wind anomalies

(contour lines, m/s) which are estimated from difference between La Niño and El Niño

628        years in the longitude band of 80°E-90°E. (c) Longitude-altitude cross-section of

temperature and zonal wind anomalies averaged between 30°N-40°N. The red and blue

lines in Figure 8b & 8c denote the tropopause altitude during La Niño and El Niño years

estimated from GNSS RO data.
