# Peer review of "Asian Summer Monsoon Anticyclone: Trends and Variability"

_Atmospheric Chemistry and Physics, 2019_

## Referee Comment (RC1) · Anonymous Referee #2 · 13 Nov 2019

This paper deals with the trends and variabilities of Asian Summer Monsoon anticyclone (ASMA) using observational and reanalysis datasets. It deals with the spatial and temporal variabilities of ASMA and its relationship with long term oscillations. The subject dealt with is a very active and relevant topic. However, as I have already pointed out in my initial review, the methodology used for the study and structure of the manuscript needs major revisions. Main concerns about the manuscript are as follows: (1) The authors must bring out the novelty of the study properly. Throughout the manuscript, the already known facts and the results of the present study are in a completely messed up state, for example in the abstract itself. The authors have stated the known facts of ASMA in the abstract. The abstract should focus on the major results of the present study. (2) In the trend analysis, the relevance of dividing the ASMA

region into four different sectors is not clear. (3) What is the sanctity in averaging the wind, when the wind magnitudes are highly inhomogeneous (calm wind near to the centre of ASMA and higher wind to the edges) in all these sectors? Spatial extent of ASMA is discussed in the manuscript. No mention about the altitude/vertical extent of ASMA. This needs to be discussed. (4) The study delineates that there is significant trend/difference in the ASMA during different decades during the period 1950-2016. The study period of ASMA variability shown in Figures 6, 7, and 8 are not clear (for active/break days, strong/weak monsoon years, and El Nino/La Nino years). Is it during the period 1948- 2016. The period of wind anomalies and temperature anomalies are not clear from the figure caption (from CHAMP and COSMIC). I think it is better to compare the variabilities for the same period. The ASMA variability for the same period as that of the COSMIC and CHAMP data can be looked into. If already, it is done so, fine. However, this is not clear from the description of the figure caption and in the text. (5) I understand that this manuscript is a part of a special issue 'Interactions between aerosols and the South West Asian monsoon'. However, this aspect is not much discussed in the manuscript. It would be nice if the authors can focus more on it.

Specific comments: Abstract: Page1 Line 9: 'These pollutants. . .'-restructure the sentence ——Line11-12: 'The pollutants are expected to make a large radiative forcing'— Name the pollutants (species) responsible for large magnitude of radiative forcing — -Line13: long term oscillations such as ....... ——-Line 19-20: 'Significant..........of the ASMA'. Significant decadal variability is observed with reference to 1951-1960 period. Restructure the sentence. ——Line 21: 'Drastic increase from westerly to easterly'- What does the sentence really mean? (later in Section 3 in Figure 5, it is seen that anomalies are obtained by removing the mean and strength is obtained by taking the difference of winds at difference latitudinal sectors. In that case is it possible to call the change of sign in anomalies as westerlies or easterlies?)

Introduction Page 2 line 39: "distant maxima characters" Is it a typo error. Did you mean "distinct maxima characteristics" ——line 41: 'The maximum occurs due to

strong winds? Rewrite this sentence. The wind in the core of the anticyclone are not strong. Distinct maxima in tracers discussed in next paragraph also. Hence, authors may combine the sentences in first and second paragraph of section 1. ——line 46: modify the word 'issue' find another suitable word ——line 50: 'confined tracers transported outside—'- What does this sentence mean? Page 3 line 53: delete 's' of Plateaus

Data and methodology Page 3 line 71: delete 's' of Centres Section 2.2: Line 89 & 93: Specify the real data period used. Whether it is 1901-2016 or 1948-2016?

Results and discussions Figure 3: caption is missing. What is the confidence level of the trend shown? Compared to the trends in the northern end, trend in the southern edge seems to be very feeble? Page 8 Line185: Is it the time series of area/spatial average of zonal wind anomalies? . . . . . . . . .Line 192: 'contaminated. . .' ? ———Line 193-194: 'One is located. . .....and the other in the ......' Rewrite the sentence Figure 5e: Why sector 30°-40°is used. This region doesn't really represent the anticyclone according to figure 4. Page 8, Figure 5: Throughout the trend analysis section, the 'shift towards westerlies'. Whether the wind is becoming westerly or becoming less easterly (ie, the strength of the easterly is reduced). Whether it is really describing the strength of the anticyclone. In Figure 5, anomalies are obtained by removing the mean and strength is obtained by taking the difference of winds at difference sectors. In that case is it possible to call the change of sign in anomalies as westerlies or easterlies ?) Page 9 Line 204: remove the bracket before 'during' Page 9 Line 221-222: whether easterly wind corresponds to cooler regions? Correct the sentence Page 10 Line 237: 'Further, . . . . . .'rewrite the sentence Page 10 Line 240: In figure 6 the blue doesn't seem to be weak. The red and blue, strength are same but opposite in direction? Page 10, Line 242: Check this sentence for the correctness of "right (left) side of the anticyclone Page 12 Lines 278-281: Check the figures and conclude the features seen in the figure only.

Please also note the supplement to this comment:
https://www.atmos-chem-phys-discuss.net/acp-2019-668/acp-2019-668-RC1-
supplement.pdf

---

## Referee Comment (RC2) · Anonymous Referee #3 · 27 Jan 2020

Review of the study entitled "Asian Summer Monsoon Anticyclone: Trends and Variability" by Ghouse Basha et al.

This is a quite interesting paper related to the Asian summer monsoon anticyclone (ASMA) and the title is adequate. The research topic is of scientific interest and worth to be publishable. The study deals with the temporal, spatial and long term trends in the ASMA by using reanalysis and satellite data sets. The authors investigated the decadal variation of the anticyclone region with respect to 1951-1960 base period. They noticed significant changes over the anticyclone edges. Furthermore, the authors also studied the ASMA variability with respect to the wet and dry spells of the Indian monsoon, strong and weak monsoon years, and the stronger El Nino Southern Oscillation (ENSO) years. Overall, the authors have brought out some significant

shortcomings from the study. However, I personally think that the paper still needs significant changes before the manuscript is ready for publication. Therefore, I recommend for publication in ACP with revision. I had the chance to read the comments of the Anonymous Reviewer #2 and I do share all his/her general comments. General comments 1. Abstract needs to be improved. I strongly suggest, the authors have to rewrite the entire abstract part and strictly focused on the important results obtained from the study. 2. How authors define the ASMA region? Why GPH values are considered to define the ASMA region? Other methods are also (for example potential vorticity) used by the previous researchers. Authors can stress this point and define their selection of ASMA region from the GPH values in the manuscript. 3. Why authors separated the ASMA into 4 parts? This needs to be discussed properly. 4. Conclusions part looks much generalized. The authors can provide 3 or 4 major results as point by point at the end of the conclusion part. 5. Finally, the presentation quality needs 'strong improvements'.

Specific comments: There are some numbers of language and grammar issues in the present manuscript. However, I do not mention all of them in the present review. The authors should take care of all in the revised version of the manuscript.

Line 7-16: Authors can shift these sentences into the introduction section.

Line 18-19: 'The decadal variability of the anticyclone is very large at the edges of anticyclone than at the core region' rewrite the sentence...

Line 20: change into 'to the 1951-1960 period'

Line 22: change 'anticyclone' to 'the anticyclone'

Line 29: '......and during strong La Nina years'. Remove 'during' from the sentence.

Line 30: Unclear——-'while interpreting the pollutants/trace gases in the anticyclone' Do you mean changes or variability in the trace gases? Please clarify what is meant here.

Line 35: 'from Asia to the Middle East'—— change it as 'from the Asia to Middle East'.

Line 35: Add 'The' in front of ASMA...

Line 89-93: data period '1901-2016/1948-2016'......This needs to be clarified.

Line 94-124: The vertical resolution of GNSS RO data was missed. What is the original resolution of the GNSS RO (CHAMP and COSMIC). Is it originally available at 100/200m or some interpolation is done?

Line 100-101: I doubt about the vertical resolution of 0.5-15 km? Is it correct? Authors can look on it again.

Line 112: 'The CHAMP data was available from 19 May 2001 to......' not required, delete this sentence.

Line 128-130: rewrite the sentence with clarity.

Line 132-134: not clear.... 'The spatial extent and intensity of anticyclone are greater during July compared to the intensities present during other months'. Rewrite the sentence.

Line 135: Authors can follow any one either 'Asia to the Middle East' or 'Middle East to East Asia' in the entire manuscript. ..... Authors mentioned earlier in Line 35 as 'Asia to the Middle East'.

Line 146: Authors written sometimes as 'anticyclone' sometimes as 'the anticyclone' in the entire manuscript. This needs to be solved in the entire manuscript.

Line 147: rewrite 'During the September month '

Line 150: change 'the core region of anticyclone'.... The core region of the anticyclone.

Line 159-173: The authors presented observed changes in the ASMA region during different decades. This paragraph needs some more discussion on the possible reasons for the observed changes.

Line 174-175: I couldn't find 'Figure 3' in the manuscript.

Line 199-203/Line 263-266: each sentence needs a citation….I suggest add some references to the sentences….

Line 207-226: The authors can give some scientific explanation on observed warming in the mid troposphere during active days.

Line 253: 'excited'? It means existed? Check it once.

Line257: This clearly demonstrates that a 'large scale ascent develops over the Asian monsoon region'. Incomplete sentence.

Line 258-259: Unclear. Rewrite the sentence again.

Line 273-274: 'the strongest El Niño (1958, 1966, 1973, 1983, 1988, 1992, 1998, and 2015) and La Niña (1974, 1976, 1989, 1999, 2000, 2008, and 2011) years'. How authors selected these years? The temperature anomalies shown in Figure 8 are from NCEP or GNSS RO? If GNSS RO, how many years considered for obtaining the temperature anomalies?

Line 307: change as 'reanalysis, satellite and observational data'

Line 308: rewrite the sentence

Line 309-310: unclear. 'Spatial (magnitude) of the anticyclone structure'

Line 313: use other suitable word instead of 'present'

Line 339: what is meant 'thermal adaptation' here? Check it once.

Line 341-344: unclear. Rewrite the sentence again.

Line 352-353: incomplete sentence.

Figures:

Figure 3 was missed from the present manuscript.

Rewrite the title of the Figure 4. . . '1948-2017' to ''1948-2016'. . ..

Figure captions needs to be improved with more clarity.

---

## Author Comment (AC1) · 6 Mar 2020

Replies to Reviewer #3 Comments/Suggestions

This is a quite interesting paper related to the Asian summer monsoon anticyclone (ASMA) and the title is adequate. The research topic is of scientific interest and worth to be publishable. The study deals with the temporal, spatial and long term trends in the ASMA by using reanalysis and satellite data sets. The authors investigated the decadal variation of the anticyclone region with respect to 1951-1960 base period. They noticed significant changes over the anticyclone edges. Furthermore, the authors also studied the ASMA variability with respect to the wet and dry spells of the Indian monsoon, strong and weak monsoon years, and the stronger El Nino South-

ern Oscillation (ENSO) years. Overall, the authors have brought out some significant shortcomings from the study. However, I personally think that the paper still needs significant changes before the manuscript is ready for publication. Therefore, I recommend for publication in ACP with revision. I had the chance to read the comments of the Anonymous Reviewer #2 and I do share all his/her general comments.

Reply: First of all we wish to thank the reviewer for handling this manuscript and for offering his/her constructive comments/suggestions, which improved the manuscript content significantly. In the revised version, we have taken care of the reviewers comments/suggestions and we hope the reviewer will find the revised version satisfactory. As per reviewer suggestion, the methodology part and structure of the manuscript is changed compared to previous version.

General comments 1. Abstract needs to be improved. I strongly suggest the authors have to rewrite the entire abstract part and strictly focused on the important results obtained from the study.

Reply: In the revised version of the manuscript, we have changed the abstract by focusing on main results only.

2. How authors define the ASMA region? Why GPH values are considered to define the ASMA region? Other methods are also (for example potential vorticity) used by the previous researchers. Authors can stress this point and define their selection of ASMA region from the GPH values in the manuscript.

Reply: We have mentioned clearly the reason for selecting the GPH values in this study in section 3.1 with complete details and references. 'The spatial extent of anticyclone circulation is clearly evident in the grid 15oN-45oN; 30oE- 120oE at 100 hPa level and the climatological averaged values of GPH varies from 16.5-17 km in NCEP reanalysis during 1948-2016. Using the modified potential vorticity equation, Randel et al. (2006) showed the spatial variation of anticyclone where GPH values are stationary in the range of 16.75-16.9 km. Similarly, Park et al. (2007) showed the anticyclone structure

from the strongest wind at 100 hPa through streamline function. Bian et al. (2012) reported the spatial variability of anticyclone using 16.77 km and 16.90 km in the GPH contour as the lower and the upper boundary, respectively. Thus, these empirically selected GPH values represent anticyclone boundaries. Therefore, in this present study, we have chosen the values from 16.75 to 16.9 km to investigate the spatial features of the anticyclone'.

3. Why authors separated the ASMA into 4 parts? This needs to be discussed properly.

Reply: In the revised version, we have given following reason for dividing the ASMA into 4 different regions. The spatial trend analysis of ASMA shows distinct variability throughout the region and the edges of the ASMA undergo drastic variability compared to other regions. Therefore, in order to understand the asymmetry in the anticyclone variability, we have divided the anticyclone region into 4 different sectors as shown in Figure 4 based on the peak values of GPH along longitude and latitude cross-sections.

4. Conclusions part looks much generalized. The authors can provide 3 or 4 major results as point by point at the end of the conclusion part.

Reply: During the first review when we submitted the manuscript, one of the reviewers suggested to remove point by point list of conclusions. Therefore, we have written the summary and conclusion part in a paragraph.

5. Finally, the presentation quality needs 'strong improvements'.

Reply: In the revised version of the manuscript, we have taken care of grammatical mistakes, general statements and other points raised by the both reviewers.

Specific comments: There are some numbers of language and grammar issues in the present manuscript. However, I do not mention all of them in the present review. The authors should take care of all in the revised version of the manuscript.

Reply: In the revised version of the manuscript, we have taken utmost care to reduce the typos and grammatical mistakes to the maximum possible extent.

Line 7-16: Authors can shift these sentences into the introduction section.

Reply: As per reviewer suggestion, we have sifted some of these lines to the introduction section.

Line 18-19: 'The decadal variability of the anticyclone is very large at the edges of anticyclone than at the core region' rewrite the sentence. . .

Reply: In the revised version of the manuscript, we have rewritten this sentence as 'Significant decadal variability is observed in the northeast and southwest parts of ASMA with reference to the 1951-1960 period'

Line 20: change into 'to the 1951-1960 period'

Reply: Changed.

Line 22: change 'anticyclone' to 'the anticyclone'

Reply: Changed.

Line 29: '. . .. . .and during strong La Nina years'. Remove 'during' from the sentence.

Reply: Removed.

Line 30: Unclear——'while interpreting the pollutants/trace gases in the anticyclone' Do you mean changes or variability in the trace gases? Please clarify what is meant here.

Reply: Written clearly in the revised version of the manuscript as 'It is suggested to consider different phases of monsoon while interpreting the variability of pollutants/trace gases in the anticyclone'

Line 35: 'from Asia to the Middle East'—– change it as 'from the Asia to Middle East'.

Reply: Changed.

Line 35: Add 'The' in front of ASMA. . .

Reply: Added.

Line 89-93: data period '1901-2016/1948-2016'. . .. . .This needs to be clarified.

Reply: Changed the year as per reviewer suggestion. The whole work is done for the period 1951-2016.

Line 94-124: The vertical resolution of GNSS RO data was missed. What is the original resolution of the GNSS RO (CHAMP and COSMIC). Is it originally available at 100/200m or some interpolation is done?

Reply: We have interpolated the data to 200m resolution and added in text.

Line 100-101: I doubt about the vertical resolution of 0.5-15 km? Is it correct? Authors can look on it again.

Reply: Sorry for this mistake. We have changed this in the revised manuscript as 'The temperature profiles from this technique are available with low horizontal ($\sim$200-300 km) and high vertical resolutions (10-35 km) with an accuracy of <0.5 K'

Line 112: 'The CHAMP data was available from 19 May 2001 to. . .. . .' not required, delete this sentence.

Reply: Deleted.

Line 128-130: rewrite the sentence with clarity.

Reply: Rewritten in the revised manuscript.

Line 132-134: not clear. . .. 'The spatial extent and intensity of anticyclone are greater during July compared to the intensities present during other months'. Rewrite the sentence.

Reply: Rewritten in the revised manuscript.

Line 135: Authors can follow any one either 'Asia to the Middle East' or 'Middle East to East Asia' in the entire manuscript. . . ... Authors mentioned earlier in Line 35 as 'Asia

to the Middle East'.

Reply: Thank you for your suggestion. We have followed Asia to Middle East throughout the manuscript.

Line 146: Authors written sometimes as 'anticyclone' sometimes as 'the anticyclone' in the entire manuscript. This needs to be solved in the entire manuscript.

Reply: Changed to 'the anticyclone'.

Line 147: rewrite 'During the September month '

Reply: Rewritten in the revised manuscript.

Line 150: change 'the core region of anticyclone'. . .. The core region of the anticyclone.

Reply: Changed.

Line 159-173: The authors presented observed changes in the ASMA region during different decades. This paragraph needs some more discussion on the possible reasons for the observed changes.

Reply: In the revised version of the manuscript, we have added more discussion as per reviewer suggestion.

Line 174-175: I couldn't find 'Figure 3' in the manuscript.

Reply: Figure 3 was merged with Figure 2 in the previous version. However, in the revised manuscript, we have added this.

Line 199-203/Line 263-266: each sentence needs a citation. . ..I suggest add some references to the sentences. . ..

Reply: References added.

Line 253: 'excited'? It means existed? Check it once.

Reply: It should be exited.

Line257: This clearly demonstrates that a 'large scale ascent develops over the Asian monsoon region'. Incomplete sentence.

Reply: Modified in the revised version as 'This process clearly demonstrates that a large scale ascent develops over the Asian monsoon region'

Line 258-259: Unclear. Rewrite the sentence again.

Reply: This sentence is edited in the revised version as 'The transport processes from the boundary layer to the tropopause occur on the east side of the anticyclone i.e. southern flank of Tibetan Plateau, northeast India and the head of the Bay of Bengal'

Line 273-274: 'the strongest El Niño (1958, 1966, 1973, 1983, 1988, 1992, 1998, and 2015) and La Niña (1974, 1976, 1989, 1999, 2000, 2008, and 2011) years'. How authors selected these years? The temperature anomalies shown in Figure 8 are from NCEP or GNSS RO? If GNSS RO, how many years considered for obtaining the temperature anomalies?

Reply: We have chosen the strong ENSO years from the website (https://ggweather.com/enso/oni.htm). The background temperatures anomalies are shown in the Figure are from NCEP reanalysis data from 1951-2016. We have used only tropopause height data from GPSRO in Figure6, 7, and 8.

Line 307: change as 'reanalysis, satellite and observational data'

Reply: Changed.

Line 308: rewrite the sentence

Reply: Rewritten in the revised manuscript.

Line 309-310: unclear. 'Spatial (magnitude) of the anticyclone structure'

Reply: The spatial extent and intensity of the anticyclone is large during July compared

to June and August.

Figures: Figure 3 was missed from the present manuscript.

Reply: Actually it was merged with figure 2. In the revised version of manuscript, we have added Figure 3 separately.

Rewrite the title of the Figure 4. . . '1948-2017' to ''1948-2016'. . ..

Reply: In the revised version, we have written clearly.

Figure captions needs to be improved with more clarity

Reply: In the revised version, figure captions are written in more elaborate way.

Once again, we would like to thank the reviewer for his/her thoughtful comments and suggestions that led to substantial improvements in the revised manuscript.

—END—

---

## Author Comment (AC2) · 6 Mar 2020

Replies to Reviewer 2 Comments/Suggestions

This paper deals with the trends and variabilities of Asian Summer Monsoon anticyclone (ASMA) using observational and reanalysis datasets. It deals with the spatial and temporal variabilities of ASMA and its relationship with long term oscillations. The subject dealt with is a very active and relevant topic. However, as I have already pointed out in my initial review, the methodology used for the study and structure of the manuscript needs major revisions.

Reply: First of all we wish to thank the reviewer for handling this manuscript and for offering his/her constructive comments/suggestions, which improved the manuscript

content significantly. In the revised version, we have taken care of the reviewers comments/suggestions and we hope the reviewer will find the revised version satisfactory. As per reviewer suggestion, the methodology part and structure of the manuscript is changed according to results.

(1) The authors must bring out the novelty of the study properly. Throughout the manuscript, the already known facts and the results of the present study are in a completely messed up state, for example in the abstract itself. The authors have stated the known facts of ASMA in the abstract. The abstract should focus on the major results of the present study.

Reply: As per reviewer suggestion, we have removed the basic introduction part from the abstract and focused on the mentioning major results of the study.

(2) In the trend analysis, the relevance of dividing the ASMA region into four different sectors is not clear.

Reply: In the revised version, we have given following reason for dividing the ASMA into 4 different regions. The spatial trend analysis of ASMA shows distinct variability throughout the region and the edges of the ASMA undergo drastic variability compared to other regions. Therefore, in order to understand the asymmetry in the anticyclone variability, we have divided the anticyclone region into 4 different sectors as shown in Figure 4 based on the peak values of GPH along longitude and latitude cross-sections.

(3) What is the sanctity in averaging the wind, when the wind magnitudes are highly inhomogeneous (calm wind near to the centre of ASMA and higher wind to the edges) in all these sectors? Spatial extent of ASMA is discussed in the manuscript. No mention about the altitude/vertical extent of ASMA. This needs to be discussed.

Reply: As mentioned above, depending upon the spatial variability of ASMA, we have divided into 4 different sectors. In order to verify the zonal wind variability in ASMA region, we have selected 3 different locations in Figure R1. At these locations also the

**ACPD**
zonal wind anomalies shows significant increasing trend similar to the variability in four different sectors. Figure R2 illustrates the vertical cross section of GPH at these three different locations (Yellow stars). From this figure it is clear that vertical extent is difficult to obtain using fixed GPH. We have also obtained composite mean spatial distribution of GPH and wind vectors at different pressure levels during different months to verify the vertical structure of ASMA (Figure R3). The wind vectors also show anticyclone structure at 200 hPa and 150hPa. However, the spatial structure of GPH extends south at 150 hPa and 200 hPa conversely shifts towards north at 70 hPa and 50 hPa. Clear spatial structure of ASMA can be visible at 100 hPa only.

(4) The study delineates that there is significant trend/difference in the ASMA during different decades during the period 1950-2016. The study period of ASMA variability shown in Figures 6, 7, and 8 are not clear (for active/break days, strong/weak monsoon years, and El Nino/La Nino years). Is it during the period 1948- 2016. The period of wind anomalies and temperature anomalies are not clear from the figure caption (from CHAMP and COSMIC). I think it is better to compare the variabilities for the same period. The ASMA variability for the same period as that of the COSMIC and CHAMP data can be looked into. If already, it is done so, fine. However, this is not clear from the description of the figure caption and in the text.

Reply: In the revised version of the manuscript, we have considered the NCEP reanalysis (wind and temperature) data from 1951 onwards only for Figure 6, 7, 8. Only the tropopause which is shown in Figure 6, 7 and 8 are derived from GPSRO (CHAMP and COSMIC) satellite data which is available from the year 2002 onward only. Compared to previous and other existing data sets, this has the highest resolution and accuary in the UTLS region at present. Note that major features will not change by choosing different time period in this aspect.

(5) I understand that this manuscript is a part of a special issue 'Interactions between aerosols and the South West Asian monsoon'. However, this aspect is not much discussed in the manuscript. It would be nice if the authors can focus more on it.
Reply: In the revised manuscript we have mentioned it clearly in the introduction and discussion section regarding importance of aerosols and trace gases variability in ASMA. Since the half part of the manuscript discusses the influence of Indian summer monsoon, we have submitted the manuscript to this special issue. Further, we have discussed the variability in trace gases and aerosols in ASMA and their relation with tropopause parameters in a separate paper which is also in ACPD.' Basha, G., Ratnam, M. V., Kishore, P., Ravindrababu, S., and Velicogna, I.: Influence of Asian Summer Monsoon Anticyclone on the Trace gases and Aerosols over Indian region, Atmos. Chem. Phys. Discuss., https://doi.org/10.5194/acp-2019-743, in review. This aspect also we have included in the revised version.

Specific comments: Abstract: Page1 Line 9: 'These pollutants. . .'-restructure the sentence  $\hat{a}\check{A}\check{T}$

Reply: Modified.

–Line11-12: 'The pollutants are expected to make a large radiative forcing'— Name the pollutants (species) responsible for large magnitude of radiative forcing —

Reply: Surface pollutants such as (CO, CH3CI). As per reviewer suggestion, the basic introduction lines are removed. The same was suggested by another reviewer.

-Line13: long term oscillations such as ......

Reply: Long term oscillations are QBO and ENSO.

—-Line 19-20: 'Significant.....of the ASMA'. Significant decadal variability is observed with reference to 1951-1960 period. Restructure the sentence. —–

Reply: The sentence is reframed as 'Significant decadal variability is observed in the northeast and southwest parts of ASMA with reference to the 1951-1960 period'.

Line 21: 'Drastic increase from westerly to easterly'- What does the sentence really mean? (later in Section 3 in Figure 5, it is seen that anomalies are obtained by remov-
ing the mean and strength is obtained by taking the difference of winds at difference latitudinal sectors. In that case is it possible to call the change of sign in anomalies as westerlies or easterlies?)

Reply: Thank you for rising this point. In the revised version we have mentioned it as anomalies.

Introduction Page 2 line 39: "distant maxima characters" Is it a typo error. Did you mean "distinct maxima characteristics" ——

Reply: Yes. We have changed in the revised version as 'distinct maximum characteristics'

line 41: 'The maximum occurs due to strong winds? Rewrite this sentence. The wind in the core of the anticyclone are not strong. Distinct maxima in tracers discussed in next paragraph also. Hence, authors may combine the sentences in first and second paragraph of section 1.

Reply: We have deleted this sentence in the revised version to avoid confusion and combined both the paragraphs.

line 46: modify the word 'issue' find another suitable word ——

Reply: Replaced 'issue' with 'problem'

line 50: 'confined tracers transported outside—'- What does this sentence mean?

Reply: We have modified the sentence in the revised version as 'The tracers which are transported are confined in the anticyclone will affect the trace gas concentration in the UTLS resulting in significant changes in radiative forcings (Solomon et al., 2010; Riese et al., 2012; Hossaini et al., 2015)'

Page 3 line 53: delete 's' of Plateaus

Reply: Deleted.

**ACPD**
Data and methodology Page 3 line 71: delete 's' of Centres

Reply: Deleted.

Section 2.2: Line 89 93: Specify the real data period used. Whether it is 1901-2016 or 1948-2016?

Reply: The whole analysis was done from the 1951 to 2016. This was clearly mentioned in manuscript.

Results and discussions Figure 3: caption is missing. What is the confidence level of the trend shown? Compared to the trends in the northern end, trend in the southern edge seems to be very feeble?

Reply: Figure caption was overlapped with figure 3. In the revised version of the manuscript, we made it visible. Trends were estimated by using robust regression analysis at 95

Page 8 Line185: Is it the time series of area/spatial average of zonal wind anomalies?

Reply: Yes. The wording is changed in the revised version.

Line 192: 'contaminated. . .' ? ———

Reply: We have re-written this sentence with better clarity in the revised manuscript.

Line 193-194: 'One is located. . ....and the other in the ..... Reply: These sentences were removed in the revised version as it is creating confusion to the reader.

Rewrite the sentence Figure 5e: Why sector 30âUe -40âUe is used. This region doesn't really represent the anticyclone according to figure 4.

Reply: We have followed the same procedure as reported Yuan et al., (2019) for estimating the strength of the anticyclone as difference in zonal winds between northern (30 o -400 N) and southern (10 o -20 oN) flanks of the ASMA. Yuan, C., Lau,

ACPD
W. K. M., Li, Z., and Cribb, M.: Relationship between Asian monsoon strength and transport of surface aerosols to the Asian Tropopause Aerosol Layer (ATAL): interannual variability and decadal changes, Atmos. Chem. Phys., 19, 1901–1913, https://doi.org/10.5194/acp-19-1901-2019, 2019. However as per reviewer suggestion, we have estimated the strength of the ASMA by taking difference in zonal wind between northern (22.5 oN-40oN) and southern (10oN-20oN). The modified figure is added in the revised version. It should be noted that, significant difference is not observed.

Page 8, Figure 5: Throughout the trend analysis section, the 'shift towards westerlies'. Whether the wind is becoming westerly or becoming less easterly (ie, the strength of the easterly is reduced). Whether it is really describing the strength of the anticyclone.

Reply: We are sorry for this mistake. In the revised version, we have mentioned it has zonal wind anomalies instead of easterly or westerly. The strength of the anticyclone was described by Yuan et al (2019) and similar method have been followed in the present study.

In Figure 5, anomalies are obtained by removing the mean and strength is obtained by taking the difference of winds at difference sectors. In that case is it possible to call the change of sign in anomalies as westerlies or easterlies ?)

Reply: Thank you for raising this important point. In the revised version, we used zonal wind anomalies only.

Page 9 Line 204: remove the bracket before 'during'

Reply: Removed.

Page 9 Line 221-222: whether easterly wind corresponds to cooler regions? Correct the sentence

Reply: Easterly wind corresponds to warmer region. This is corrected in the revised version.

**ACPD**
Reply: This sentence is modified in the revised version as 'The composite of mean distribution of the anticyclone circulation during strong and weak monsoon years is shown in Figure 7a based on GPH values at 100 hPa from NCEP reanalysis data'

Page 10 Line 240: In figure 6 the blue doesn't seem to be weak. The red and blue, strength are same but opposite in direction

Reply: This sentence is changed in the revised version of the manuscript as 'The circulation expands on the eastern and western sides of the anticyclone during the strong monsoon years (red line)'

Page 10, Line 242: Check this sentence for the correctness of "right (left) side of the anticyclone

Reply: This is corrected in the revised version.

Page 12 Lines 278-281: Check the figures and conclude the features seen in the figure only

Reply: Thank you for your suggestion. We have edited in the revised version.

Once again, we would like to thank the reviewer for his/her thoughtful comments and suggestions that led to substantial improvements in the revised manuscript.

-END-
**ACPD**
**Fig. 1.** Figure R1. (Top) Climatology of GPH showing ASMA. (Bottom) Time series of zonal wind anomalies at different locations (mentioned with yellow stars) in ASMA region from NCEP reanalysis data during 1951

**ACPD**
Fig. 2. Figure R2. Vertical cross section of GPH at three different locations

---

## Author Response (AR2)

**Replies to Editor # Comments/Suggestions**

Dear authors, please check the wording and also citations carefully. I have found the phrase 'well-known' , e.g. page 11 line 262 but no citations. Please go through your manuscript and check if you backed all claims with citations. It has been a major criticism that you mixed up known results and results obtained by you.

**Reply: First of all, we wish to thank the Editor for handling the manuscript and for offering his/her constructive comments/suggestions, which improved the manuscript content significantly. In the revised version, we have taken care of proper citation to previous and differentiated the results.**

On page 11 line 275: what do you mean here? La Nina is a an atmospheric condition. How can El Nino extend over El Nina (I suppose you mean La Nina?)?
**Reply: we have revised the this statement in the manuscript as 'During the La Niña, the anticyclone circulation extends large compared to El Niño years at 100 hPa as shown in the Figure 8a'**

On page 8. The phrases 'good rainfall' and 'less rainfall' sound rather subjective. I think you should briefly summarize the definition of active and break days and not force the reader to study another paper.

**Reply: we have added the detailed discussion about the definition of active and break in the revised version.**

**Once again, we would like to thank the Editor for his/her thoughtful comments and suggestions that led to substantial improvements in the revised manuscript.**

---END---

[revised manuscript text omitted]